# Lesions in a songbird vocal circuit increase variability in song syntax

Avani Koparkar[1†], Timothy L Warren[2,3†], Jonathan D Charlesworth[2], Sooyoon Shin[2], Michael S Brainard[2], Lena Veit[1]*

[1]Neurobiology of Vocal Communication, Institute for Neurobiology, University of Tübingen, Tübingen, Germany; [2]Howard Hughes Medical Institute and Center for Integrative Neuroscience, University of California San Francisco, San Francisco, United States; [3]Departments of Horticulture and Integrative Biology, Oregon State University, Corvallis, United States

**Abstract** Complex skills like speech and dance are composed of ordered sequences of simpler elements, but the neuronal basis for the syntactic ordering of actions is poorly understood. Bird-song is a learned vocal behavior composed of syntactically ordered syllables, controlled in part by the songbird premotor nucleus HVC (proper name). Here, we test whether one of HVC's recurrent inputs, mMAN (medial magnocellular nucleus of the anterior nidopallium), contributes to sequencing in adult male Bengalese finches (*Lonchura striata domestica*). Bengalese finch song includes several patterns: (1) *chunks,* comprising stereotyped syllable sequences; (2) *branch points*, where a given syllable can be followed probabilistically by multiple syllables; and (3) *repeat phrases*, where individual syllables are repeated variable numbers of times. We found that following bilateral lesions of mMAN, acoustic structure of syllables remained largely intact, but sequencing became more variable, as evidenced by 'breaks' in previously stereotyped chunks, increased uncertainty at branch points, and increased variability in repeat numbers. Our results show that mMAN contributes to the variable sequencing of vocal elements in Bengalese finch song and demonstrate the influence of recurrent projections to HVC. Furthermore, they highlight the utility of species with complex syntax in investigating neuronal control of ordered sequences.

**\*For correspondence:**
lena.veit@uni-tuebingen.de

[†]These authors contributed equally to this work

## eLife assessment

Songbirds provide a tractable model system to study mechanisms of vocal production and sequencing, and past work showed that lesions to lMAN, the output of a basal ganglia thalamo-cortical loop, reduced vocal variability, consistent with a role in motor exploration. In this **fundamental** work, the authors rigorously examined how lesions to an understudied neighboring region, mMAN, part of a parallel basal ganglia loop, affect singing in Bengalese finches, whose songs exhibit complex sequential transitions. The authors provide **compelling** evidence that mMAN lesions resulted in increased sequential variability but do not affect syllable acoustic structure, showing that distinct frontal systems can have distinct functions for producing and sequencing song syllables.

## Introduction

Complex behaviors are composed of sequences of simpler motor elements. For example, in human speech, sentences are formed by syllables that are flexibly sequenced according to syntactic rules (*Berwick et al., 2011*), which define the order of individual elements. Production of motor sequences is important not only for speech, but also for most other natural movements. Despite the ubiquity of syntactically organized sequential behaviors, little is known about how the brain produces flexible

motor sequences (*Lashley, 1951*; *Tanji, 2001*; *Aldridge and Berridge, 2003*; *Wiltschko et al., 2015*; *Veit et al., 2021*).

Birdsong is a learned behavior that is well suited for the study of the neural control of motor sequences. Song is composed of discrete elements, called syllables, that are organized according to syntactic rules. The complexity of song syntax varies widely across different songbird species – from the simple and repetitive songs of owl finches (*Wang et al., 2019*) and zebra finches (*Zann, 1996*) to the variable songs of Bengalese finches (*Honda and Okanoya, 1999*) and canaries (*Nottebohm et al., 1976*; *Cohen et al., 2020*) up to the immense repertoire of nightingales (*Hultsch et al., 2004*; *Costalunga et al., 2023*). The neuronal mechanisms of birdsong production have been primarily studied in the zebra finch, partly by leveraging the structure of zebra finch song, in which syllables are produced in a relatively stereotyped and linear sequence. Consequently, we have a detailed understanding of the neuronal mechanisms for producing individual song syllables in this species, but we know little about how syllables are organized into the more complex and variable sequences that are characteristic of many other species (*Ivanitskii and Marova, 2022*).

Song production is controlled by neuronal activity in a set of brain nuclei called the song motor pathway, including premotor nucleus HVC (proper name). HVC neurons projecting to primary motor nucleus RA (*robust nucleus of the arcopallium*) encode syllable identity and have been shown to play a role in controlling syllable timing (*Simpson and Vicario, 1990*; *Vu et al., 1994*; *Hahnloser et al., 2002*; *Long and Fee, 2008*; *Aronov et al., 2011*; *Ölveczky et al., 2011*; *Lynch et al., 2016*; *Picardo et al., 2016*; *Zhang et al., 2017*). HVC activity in Bengalese finches (*Sakata and Brainard, 2008*; *Fujimoto et al., 2011*) and canaries (*Cohen et al., 2020*) has also been shown to encode syllable identity and can additionally influence sequencing and reflect the sequential context in which syllables are produced (*Fujimoto et al., 2011*; *Zhang et al., 2017*; *Cohen et al., 2020*). The nuclei of the motor pathway are essential for song production (*Nottebohm et al., 1976*; *Scharff and Nottebohm, 1991*), but the extent to which these sequence-selective activity patterns in HVC reflect internal neural dynamics or are shaped by recurrent inputs is poorly understood.

Although HVC is often considered to be at the top of a hierarchy in the song control circuit, it is part of a network of brain nuclei involved in song production and receives multiple inputs that might also contribute to premotor activity (*McCasland, 1987*; *Hosino and Okanoya, 2000*; *Jin, 2009*; *Hamaguchi et al., 2016*; *Vyssotski et al., 2016*). One prominent source of input to HVC is mMAN (*medial magnocellular nucleus of the anterior nidopallium*), which is part of a recurrent loop that bilaterally connects the two hemispheres through RA and bilateral connections from thalamic nucleus DMP (*dorsomedial nucleus of the posterior thalamus, Figure 1E*) and could thus be ideally positioned to convey sequence-related activity across hemispheres (*Vates et al., 1997*; *Schmidt et al., 2004*; *Williams et al., 2012*). A parallel recurrent pathway, which projects to RA through nucleus lMAN (*lateral magnocellular nucleus of the anterior nidopallium, Figure 1E*) has been shown to contribute variability to syllable pitch, a parameter that is controlled by RA premotor activity (*Kao et al., 2005*; *Sober et al., 2008*; *Miller et al., 2017*). lMAN-guided variability can serve as a form of motor exploration that is essential for learning (*Sober and Brainard, 2012*; *Dhawale et al., 2017*), regulating social context-dependent changes in pitch variability (*Kao and Brainard, 2006*; *Hampton et al., 2009*), and biasing pitch in the direction of improved motor performance during learning (*Andalman and Fee, 2009*; *Warren et al., 2012*; *Tian and Brainard, 2017*). This raises the question of whether mMAN, as the output of a parallel recurrent pathway that projects onto HVC, might have a similar role in contributing to premotor activity in its projection target, with the potential to influence syllable identity or sequencing variability (*Kubikova et al., 2007*; *Seki and Okanoya, 2008*; *Ali et al., 2013*). Previous studies in zebra finches showed that mMAN lesions in juveniles disrupt normal song learning, while lesions in adults affected song initiation, but not subsequent production of song in this species (*Foster and Bottjer, 2001*; *Horita et al., 2008*; *Ali et al., 2013*).

We tested the hypothesis that mMAN contributes to syllable sequencing by studying the effect of bilateral mMAN lesions on song production in adult Bengalese finches. We found that lesions had little effect on the acoustic structure of individual syllables but led to an increase in the variability of multiple aspects of syllable sequencing. These results highlight the potential importance of recurrent inputs such as mMAN in shaping the syntactical structure of adult birdsongs.

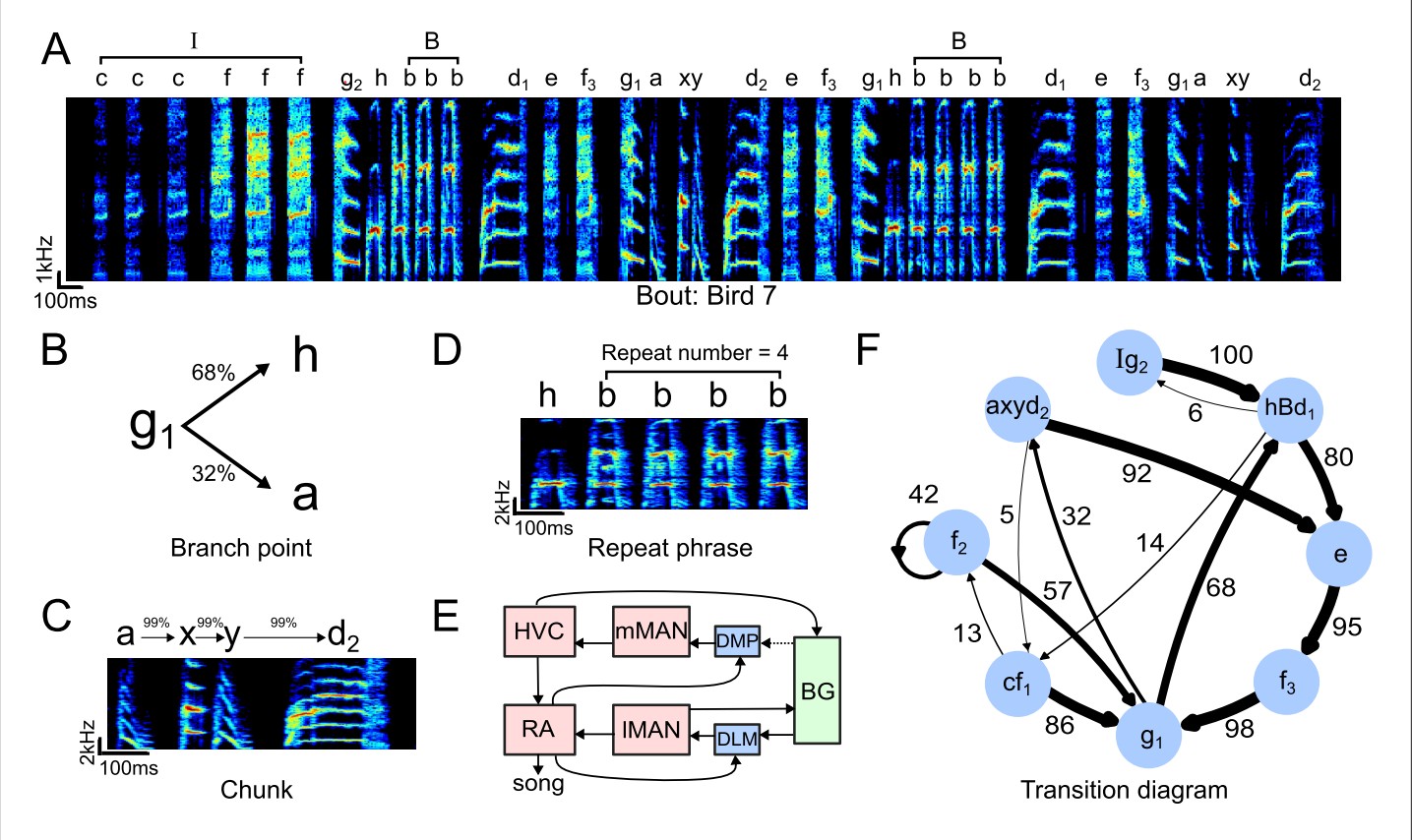

**Figure 1.** Structure of Bengalese finch song. (**A**) Example spectrogram (bird 7) depicting an entire song bout with introductory state '*I*', repeat phrase '*B*', and individual syllables. (**B**) Example transition diagram depicting a branchpoint with variable sequencing. Numbers above the arrows denote transition probabilities in percent. (**C**) Example spectrogram of a chunk. Chunks are defined as highly stereotyped sequences of syllables that only have a single input and output branch and are condensed into one state in the transition diagram (see 'Methods'). (**D**) Example spectrogram of a repeat phrase, summarized by capital letter '*B*' in the transition diagram. The repeating syllable (here, syllable '*b*') repeats a variable number of times across different instances of the repeat phrase. (**E**) Schematic showing recurrent pathways projecting onto motor pathway nuclei through lMAN and mMAN. Red: pallial nuclei; blue: thalamic nuclei; green: basal ganglia. Dotted line indicates suspected connection by *Kubikova et al., 2007*. mMAN: medial magnocellular nucleus of the anterior nidopallium; DMP: dorsomedial nucleus of the posterior thalamus; BG: basal ganglia; DLM: medial portion of the dorsolateral thalamus; lMAN: lateral magnocellular nucleus of the anterior nidopallium; RA: robust nucleus of the arcopallium. (**F**) Example of a transition diagram. Nodes denote chunk or syllable labels, numbers denote transition probabilities (in percent, % symbol omitted for clarity), $d_1/d_2$, $g_1/g_2$ denote different states of syllables *d* and *g* respectively based on different sequential contexts, capital letters denote repeat phrases. Edges at each node may not sum to 100% because branches smaller than 5% are omitted for clarity.

The online version of this article includes the following figure supplement(s) for figure 1:

**Figure supplement 1.** Image of calcitonin gene-related peptide (CGRP)-stained frontal section (left) control and (right) bird 5.

**Figure supplement 2.** Transition entropy after medial magnocellular nucleus of the anterior nidopallium (mMAN) lesions remains elevated over several days.

## Methods

**Key resources table**

| Reagent type (species) or resource | Designation | Source or reference | Identifiers | Additional information |
|---|---|---|---|---|
| Antibody | Anti-CGRP antibody (rabbit polyclonal) | Sigma-Aldrich, St. Louis, MO; 10.1159/000113342 | | Dilution 1:10,000 |
| Software, algorithm | Custom recording software, LabView | 10.1038/nature06390 | | |
| Software, algorithm | Sound Analysis Pro software | 10.1006/anbe.1999.1416 | | |
| Software, algorithm | SoundSig sound analysis package | 10.1007/s10071-015-0933-6 | | |
| Software, algorithm | TweetyNet sound annotation package | 10.7554/eLife.63853 | | |

### Subjects and sound recordings

Experiments were carried out on seven adult male Bengalese finches (*Lonchura striata domestica*) obtained from the Brainard lab's breeding colony at University of California, San Francisco (median age 640 d post-hatch, range 149–1049 at the start of experiment). Birds were raised with their parents and then housed in same-sex group cages. For the experiments, birds were placed in individual sound-attenuating boxes (Acoustic Systems, Austin, TX) and maintained on a 14:10 hr light:dark period. Song was recorded using an omnidirectional microphone above the cage using custom LabView software for continuous monitoring of song output (*Tumer and Brainard, 2007*). All experimental work was carried out under protocols approved by the University of California, San Francisco's (UCSF) institutional animal care and use committee (IACUC); protocol entitled "The neural basis of vocal learning in songbirds," including versions AN185512, AN170723, AN107972, AN087315, AN080388, AN075783.

### mMAN lesions

After several days of sound recording, birds were anesthetized using ketamine and midazolam with isoflurane. mMAN was localized using stereotaxic coordinates and was destroyed bilaterally using current injections of 100 uA current for 80 s at four sites per hemisphere. After conclusion of the study, birds were deeply anesthetized and perfused with paraformaldehyde. 40 um sections were cut on a microtome and processed with either Nissl stain (cresyl violet) or calcitonin gene-related peptide (CGRP) staining (*Hampton et al., 2009*). mMAN lesion size was estimated by experienced observers and estimated to be either complete (bird 1, birds 3–6) or greater than 75% (birds 2 and 7) for all birds used in this analysis (*Figure 1—figure supplement 1*).

### Sequence analysis

Bengalese finch songs were recorded for several days before and after lesions. Singing activity resumed on average 3.8 d post lesions (range 2–6 d, *Figure 1—figure supplement 1*), and we analyzed songs starting on average 5.3 d post lesions (range 3–11 d) up to 7.4 d post lesions (range 5–12 d). We analyzed at least 2 d pre and post lesions, and subsampled the larger of the two datasets pooled over all days such that the pre and post datasets were equal in size for each bird (average of 299 song bouts, range 102–601, for all analyses). All our main analyses were carried out on this dataset. To analyze specifically the persistence of effects after the lesion, *Figure 1—figure supplement 2* includes four additional data points (14, 19, 33, and 33 d post lesion for birds 2, 3, 6, and 7, respectively) after additional behavioral manipulation, from which birds typically recover (*Warren et al., 2012*).

### Annotation

We annotated the entire syllable sequence in an automated way. We first used a subset of the data to generate a training set for classification. Separate classifiers were trained for each bird. In the training data, syllable onsets and offsets were determined by amplitude thresholding. We then performed dimensionality-reduction and unsupervised clustering of the spectrograms of detected syllables using UMAP (*McInnes et al., 2018*; *Sainburg et al., 2020*) in order to determine the number of different syllables in an objective way (mean 13, range 11–14). On average, we used 181 files (range 70–505) for creating this training dataset, which was then used to train a deep neural network (TweetyNet) for

the annotation of birdsong (*Cohen et al., 2022*). TweetyNet was used to segment and annotate all songs, followed by semi-automated hand-checking using custom-written software in MATLAB R2021b to ensure quality annotation. In rare cases, where there was some ambiguity in the assignment of syllable identity during hand-checking, we additionally took into account the sequential context, for example, during stereotyped chunks, to assign the most likely label. Thus, any errors in assignment in such cases would have tended to reduce rather than accentuate the magnitude of the lesions' effects on reported sequencing changes.

## Simplification of the syllable sequence

We defined song bouts as continuous sequences of syllables separated by at least 2 s of silence. We then simplified the syllable sequence by merging introductory notes into a single introductory state, that is, a single node '*I*' in the transition diagram (*Figure 1A and F*). Introductory notes were defined as up to three syllable types occurring at the start of bouts that were quieter, and more variable in timing and structure than other syllable types (*Rajan and Doupe, 2013*; *Veit et al., 2021*). Similarly, syllables that were repeated a variable number of at least two times were reduced to a single state (e.g., '*bbbbbbbbb*' to '*B*', *Figure 1A and F*) in the syllable sequence, and the number of repetitions within these repeat phrases was analyzed separately (*Hampton et al., 2009*; *Zhang et al., 2017*; *Jaffe and Brainard, 2020*). We only analyzed repeat phrases with variable numbers of syllable repetitions in this way; therefore, we did not consider cases in which a syllable was repeated an exact number of times on each occurrence (occurring with repeat numbers of 2 or 3 in our dataset, see Figure 3A) a variable repeat phrase.

## Separating different syllable states based on sequential context

Transition probabilities between syllables in Bengalese finch song can depend not only on the identity of the syllable, but also on the identity of at least one preceding syllable (*Jin and Kozhevnikov, 2011*; *Katahira et al., 2011*; *Morita et al., 2021*). In order to consider this sequential context-dependency of transition probabilities, we followed the procedure of *Katahira et al., 2011* to classify spectrally similar syllables (i.e., with the same labels) into different 'states' based on the identity of the preceding syllable. For example, in *Figure 1A*, the syllable '*g*' is preceded by '*I*' or '*f*' and is followed by '*h*' or '*a*'. If '*g*' is preceded by '*I*', it is always followed by '*h*' and if it is preceded by '*f*', it can be followed by '*h*' or '*a*'. We would therefore like to introduce two separate states $g_1$ and $g_2$ in the transition diagram representing this sequential context. The input for this analysis was the syllable sequence with introductory and repeat phrases already condensed into single states. We compared the distribution of transition probabilities from these syllables by themselves (e.g., syllable '*g*') to the distribution of transition probabilities from these syllables considering the context of one preceding syllable (e.g., syllable '*f-g*') using the chi-square goodness-of-fit test. We split the syllable into a new state (e.g., '$g_1$') if p<0.01/n, where *n* is the number of comparisons made (Bonferroni correction for multiple comparisons) (*Jin, 2009*; *Katahira et al., 2013*). After completing this for all syllables of a given type (e.g., '$g_1$', '$g_2$', '$g_3$'), we compared the different output states to each other using chi-square goodness-of-fit test and merged them back together if *p*>0.01/n (mean 15, range 6–21 additional states/bird).

## Determining chunks and branchpoints

To determine chunks (*Suge and Okanoya, 2010*), we used the modified syllable sequence after chi-square analysis. We calculated a syllable-to-syllable transition diagram, while eliminating any syllables occurring with less than a threshold frequency of 0.9% and omitting branches with a probability of less than 5%. From the resultant graph, we merged all non-overlapping linear sequences (which consisted of nodes with only a single input and output branch) into chunks. The transition probabilities within resultant chunks were typically higher than 90% (range 87–100%). Values smaller than 100% for non-branching paths result from the omission of one or several small branches of less than 5%. We then re-calculate the transition probabilities between chunked nodes. We defined branchpoints as the set of single syllables that are retained after this processing and the end syllables of the newly defined chunks. To simplify diagrams for visualization, we eliminated any syllables that occurred less than once per bout and branches with transition probabilities of less than 5%.

## Transition entropy

Transition entropy is a measure of uncertainty of sequence at a given syllable (*Sakata and Brainard, 2006*). With $c$ different outputs from the given syllable '*a*' and $P(i)$ the probability of the $i$th outcome, we calculate the entropy $H_a$ as

$$H_a = -\sum_{i=1}^{c} P(i) \, logP(i)$$

We call this value 'transition entropy per branchpoint' in Figure 3A.

To determine the overall variability of song before and after mMAN lesions, we calculated total transition entropy *TE*, over all syllables '*b*' as:

$$TE = -\sum_{b=1}^{n} H_b * P(b)$$

where $H_b$ is the transition entropy at '*b*' and $P(b)$ is the frequency of syllable '*b*' (*Chatfield and Lemon, 1970*; *Katahira et al., 2013*).

## History dependence

History dependence is a previously established metric that measures the extent to which the most common transition at a given syllable is influenced by the transition at the last occurrence of this syllable (*Warren et al., 2012*). It has been used to characterize instances of apparent sequence variability, where seemingly variable transitions are always strictly alternating. For example, if the possible transitions from syllable '*a*' are '*ab*' or '*ac*' but these strictly alternated ('*ab … ac … ab … ac*' and so on), then the seemingly variable branchpoint '*a*' is perfectly predictable based on its history (*Warren et al., 2012*). Such apparent variability should be largely eliminated in our sequence analysis by the introduction of context-dependent states (i.e., in this example, the '*a*' would be re-labeled as '$a_1$' or '$a_2$' depending on the context in which it occurs) and identification of chunks. However, if higher-order dependencies in the song determine the order of chunks, we might still expect some variable transitions to be governed by history dependence. If '*ab*' is the most frequently occurring transition from '*a*', and '*ac*' is the collection of all other transitions from '*a*', we define history dependence D of '*a*' as:

$$D = P\left(ab_n | ab_{n-1}\right) - P\left(ab_n | ac_{n-1}\right) \vee$$

where $P\left(ab_n | ab_{n-1}\right)$ is the conditional probability of '*ab*' transition given that '*ab*' transition occurred at the previous instance of '*a*' and $P\left(ab_n | ac_{n-1}\right)$ is the conditional probability of '*ab*' transition given that '*ac*' transition occurred at the previous instance of '*a*'.

## Chunk consistency

As defined above, a chunk is defined by a single dominant sequence, but may have a small amount of variability across different instances. To quantify the stereotypy of chunks, we used a measure based on sequence consistency previously defined for relatively stereotyped zebra finch songs (*Scharff and Nottebohm, 1991*). Across all instances of a given chunk, we identified the syllable sequence that occurred most often as the 'dominant sequence'. We then defined '*n_dominant*' as the number of instances of the dominant sequence, and '*n_other*' as the number of instances of other sequence variants for the chunk.

We quantified chunk consistency *C* as the proportion of total instances that were the dominant sequence:

$$C = \frac{n_{dominant}}{n_{dominant} + n_{other}}$$

To compare a chunk before and after mMAN lesions, the dominant sequence for the pre-lesion chunk was used as a reference, regardless of whether the same sequence qualified as a chunk post lesion. To quantify chunk consistency post lesion, the most dominant sequence post lesion was used (even if that was not the same as the most dominant sequence pre lesion).

### Repeat number variability

To study the influence of mMAN on repeat phrases, we examined the distribution of repeat numbers before and after lesions. We quantified the variability of these distributions as their coefficient of variation (*CV = standard deviation/mean*).

## Analysis of syllable acoustic structure

### Average spectrogram

For visual inspection of preservation of syllable types before and after mMAN lesions, we calculated average spectrograms for 200 instances of each syllable type for seven birds (*Figure 2—figure supplement 1*). Individual spectrograms of each syllable type were rescaled to have matching time axes and then overlaid.

### Syllable similarity analysis

We used Sound Analysis Pro (SAP) software (*Tchernichovski et al., 2000*) to quantify the similarity in syllable structure for all syllable types per bird (*Figure 2—figure supplement 2*). We used the settings 'Symmetric' and 'Time Courses' and compared 20 instances of each syllable type.

We first calculated SAP similarity of all syllables to syllables of the same type from two separate control recordings before mMAN lesions. We termed this as 'Self Similarity'. We next calculated SAP similarity of the same syllable types before and after mMAN lesions. We termed this as 'Pre vs Post' similarity. Lastly, we calculated SAP similarity of all syllable types compared to all other syllable types per bird. This was used as a measure of dissimilarity of structure across syllable types. We termed this as 'Cross Similarity'.

### Analysis of fundamental frequency

We used the method previously described by *Hampton et al., 2009* to calculate fundamental frequency and CV of fundamental frequency for select syllables for each bird (*Figure 2—figure supplement 3*; *Hampton et al., 2009*). Briefly, each syllable was visually inspected to have a segment with clearly defined fundamental frequency. We then found the peak of the autocorrelation function (using parabolic interpolation) of the sound waveform in this segment.

### Analysis of acoustic features

To further compare syllable phonology before and after the mMAN lesions, we used the SoundSig Python package (*Elie and Theunissen, 2016*) to calculate a set of acoustic features for all syllables for seven birds (*Figure 2—figure supplement 4*). The features used were entropy of spectral envelope (entS), temporal centroid for the temporal envelope (meanT), and first, second, and third formants (F1, F2, F3).

## Results

### Structure of Bengalese finch song

Bengalese finch songs are composed of a fixed number of syllable types, which are arranged into variable sequences following syntactical rules. In this study, we sought to first characterize the different features of Bengalese finch song before investigating their change following mMAN lesions. For each bird, we first annotated all songs with labels for each of the syllable types, and then defined a transition diagram for analysis according to the following procedure (see 'Methods'):

1. We identified introductory notes (*Rajan and Doupe, 2013*; *Veit et al., 2021*) and grouped them together into a single introductory state '*I*' in the transition diagram. This reduced the overall transition entropy of song for seven birds from 0.95 to 0.85 (*Figure 1A*).
2. Repeat phrases, in which the same syllable is repeated a variable number of times, were also reduced to a single state in the transition diagram (indicated by capital letters) and the distribution of repeat numbers was analyzed separately (*Hampton et al., 2009*; *Zhang et al., 2017*; *Jaffe and Brainard, 2020*; *Figure 1D*).
3. In Bengalese finch song, a single spectrally defined and labeled syllable can occur in different fixed sequences (e.g., syllable '*b*' in the sequences *a-b-c* and *d-b-e*) and therefore reflects

distinct states in the song sequence (*Wohlgemuth et al., 2010*; *Katahira et al., 2011*). We followed the procedure of Katahira et al. to determine if any initially identified syllable types corresponded to more than one state based on whether the preceding syllable carried information on transition probabilities (see 'Methods', *Katahira et al., 2013*). After identifying and splitting these 'hidden states' (e.g., $b_1$, $b_2$) for each bird, overall transition entropy was further reduced from 0.85 to 0.54 (*Figure 1F*).

4. In the new syllable sequence, we identified 'chunks' (*Suge and Okanoya, 2010*), sub-sequences of high-probability transitions. We defined these as paths of continuous, non-branching sequences in the transition diagram, after omitting branches that occurred with probability <5% (*Figure 1C*). Lastly, we identified branch points in the transition diagram as the set of single syllables and endpoints of chunks, that is, all states where the syllable sequence proceeds in a probabilistic manner (*Figure 1B and F*).

## Transition entropy increased after bilateral mMAN lesions

After bilateral mMAN lesions, all syllable types were preserved (*Figure 2—figure supplements 1–4*), but the sequencing of syllables became more variable. In the example in *Figure 2B and C*, we show the transition diagrams of one example bird before and after mMAN lesions. We observed a change in transition probabilities at existing branch points, as well as the appearance of novel branches. We quantified changes in sequencing variability for all syllables using transition entropy (*Sakata and Brainard, 2006*; *Katahira et al., 2013*). Total transition entropy increased significantly after mMAN lesions, indicating that song syntax overall became more variable (*Figure 2D*). These effects appeared as soon as the bird resumed singing after the lesions and did not recover over the time for which the birds were observed (*Figure 1—figure supplement 2*).

*Figure 2—figure supplement 5A* shows the corresponding changes at all individual branch points. To investigate the source of differences in the magnitude of changes across branch points, we considered the history dependence of each of these branchpoints. History dependence captures apparent variability governed by long-range dependencies in the sequence and history-dependent transitions previously have been found to be difficult to modify in a sequence modification training protocol (*Warren et al., 2012*). They might therefore be less affected by lesions as well. Alternatively, we might expect lesion effects to be stronger for these transitions if mMAN contributes specifically to long-range dependencies. Consistent with the first possibility, we observed that there was a nonsignificant trend toward larger changes after mMAN lesion for transitions with low history dependence (*Figure 2—figure supplement 5B*).

## Chunks became more variable after bilateral mMAN lesions

Bengalese finch songs contain short, relatively stereotyped sequences of syllables, here called 'chunks' (*Okanoya and Yamaguchi, 1997*; *Suge and Okanoya, 2010*; *Isola et al., 2020*; *Veit et al., 2021*). We speculated that transitions within chunks might be differently affected by mMAN lesions if, for example, the relatively fixed sequences within chunks are determined within premotor song nucleus HVC and inputs to HVC are only relevant at variable branch points. We found that the relatively fixed sequences within chunks were altered after mMAN lesions. An example bird (*Figure 3A–C*) exhibited one prominent chunk that was occasionally observed to 'break' after lesions, that is, branching within the chunk increased to a degree where it would no longer be considered a chunk after the lesions. The branching increased most noticeably in the 'l-f' transition (*Figure 3C*), which is characterized by a longer gap than other transitions within the chunk (*Figure 3A*). We therefore wondered whether sequencing changes were related to gap durations and found that gap durations within chunks were not significantly correlated with the increase in transition entropy at the corresponding transitions (*Figure 3—figure supplement 1A*). Overall, we found that changes within chunks were approximately the same magnitude as changes at branch points (*Figure 3—figure supplement 1B*).

We quantified changes in within-chunk transitions using sequence consistency (*Scharff and Nottebohm, 1991*), a measure previously used to describe the relatively consistent sequence of zebra finch song. Sequence consistency measures how consistently the most probable sequence is followed (see 'Methods'). Sequence consistency within chunks significantly decreased across 23 chunks from all birds, indicating that this aspect of song structure was affected by mMAN lesions (*Figure 3D*).

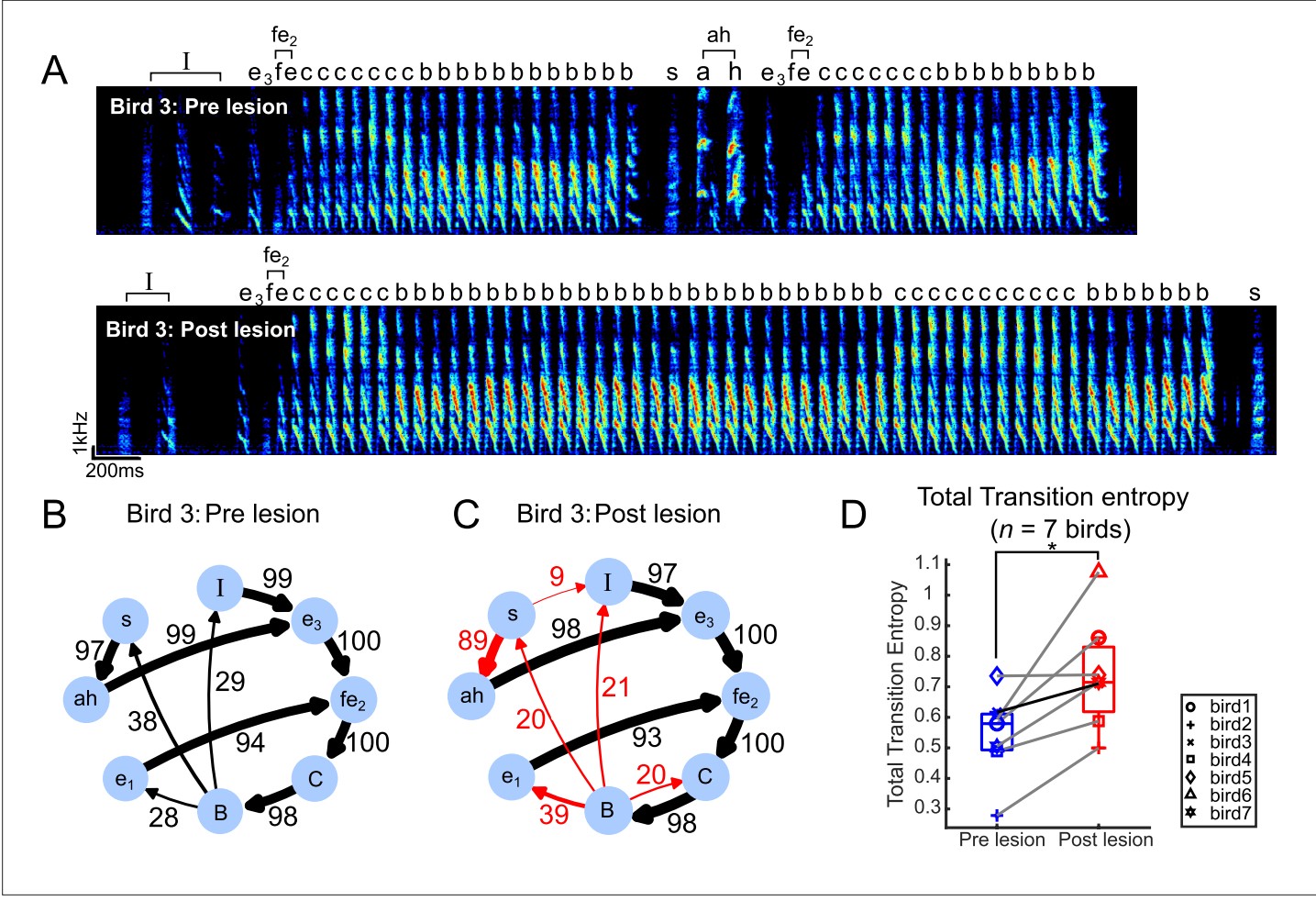

**Figure 2.** Transition entropy increased after bilateral medial magnocellular nucleus of the anterior nidopallium (mMAN) lesions. (**A**) Example spectrogram (bird 3) pre lesion (above) and post lesion (below). '*I*' denotes introductory state, and '*fe₂*' and '*ah*' denote chunks, shown as single nodes in the transition diagram. (**B**) Pre-lesion transition diagram, as in **Figure 1F**. Note that '*fe₂CB*' is also a chunk before the lesions but is shown as three separate nodes in order to align with the post-lesion diagram in (**C**). (**C**) Post-lesion transition diagram. Arrows in red mark example nodes with relatively high increase in transition entropy, including the introduction of new branches after mMAN lesions. (**D**) Total transition entropy for seven birds before and after mMAN lesions (*p<0.05, n = 7, Wilcoxon signed-rank test). Example bird from (**A–C**) is shown as darker line. Boxes indicate interquartile range and whiskers mark data points within one additional interquartile range.

The online version of this article includes the following source data, source code, and figure supplement(s) for figure 2:

**Source code 1.** Code for generating **Figure 2D** based on data from **Source data 1–28**.

**Figure supplement 1.** Average spectrograms of 200 instances of all syllable types for all birds before and after medial magnocellular nucleus of the anterior nidopallium (mMAN) lesions.

**Figure supplement 2.** Syllable similarity calculated using Sound Analysis Pro (SAP).

**Figure supplement 3.** Changes in pitch variation following mMAN lesions to control for lMAN damage.

**Figure supplement 4.** Selected acoustic features for all syllables in all birds before and after medial magnocellular nucleus of the anterior nidopallium (mMAN) lesions.

**Figure supplement 4—source data 1.** Spectral features data for generating **Figure 2—figure supplement 4** for bird 1.

**Figure supplement 4—source data 2.** Spectral features data for generating **Figure 2—figure supplement 4** for bird 2.

**Figure supplement 4—source data 3.** Spectral features data for generating **Figure 2—figure supplement 4** for bird 3.

**Figure supplement 4—source data 4.** Spectral features data for generating **Figure 2—figure supplement 4** for bird 4.

**Figure supplement 4—source data 5.** Spectral features data for generating **Figure 2—figure supplement 4** for bird 5.

**Figure supplement 4—source data 6.** Spectral features data for generating **Figure 2—figure supplement 4** for bird 6.

**Figure supplement 4—source data 7.** Spectral features data for generating **Figure 2—figure supplement 4** for bird 7.

*Figure 2 continued on next page*

*Figure 2 continued*

**Figure supplement 4—source code 1.** Code for generating *Figure 2—figure supplement 4* based on data from *Figure 2—figure supplement 4—source data 1*.

**Figure supplement 5.** Change in transition entropy for branchpoints.

**Figure supplement 5—source code 1.** Code for generating *Figure 2—figure supplement 5B* based on data from *Source data 1 - Source data 28*.

**Figure supplement 5—source code 2.** Code for generating *Figure 2—figure supplement 5A* based on data from *Source data 1 - Source data 28*.

## Repeat numbers became more variable after mMAN lesions

We next tested how mMAN lesions affected repeat phrases. Repeat phrases might be governed by separate neural mechanisms than branch points (*Fujimoto et al., 2011*; *Jin and Kozhevnikov, 2011*; *Wittenbach et al., 2015*; *Zhang et al., 2017*). In our dataset of seven birds, only five birds had songs which contained repeat phrases.

In the example bird in *Figure 4A and B*, the average repeat number for the indicated syllable increased after mMAN lesions. Across birds, the mean repeat number did not change consistently, although individual birds could exhibit quite dramatic effects on repeat number (*Figure 4B and C*; p>0.05, n = 6, Wilcoxon signed-rank test). The distributions of repeat numbers became wider and the coefficient of variation for repeat numbers increased significantly after mMAN lesions (*Figure 4D*; *p<0.05, n = 6, Wilcoxon signed-rank test).

## Discussion

Birdsongs are vocal sequences composed of individual syllables that appear in sequential order dictated by syntactic rules, but the neuronal control of song syntax is largely unknown. Previous studies in zebra finches showed that mMAN lesions in adult birds had minimal effects on song production (*Foster and Bottjer, 2001*; *Horita et al., 2008*; *Ali et al., 2013*). We speculated that the modest effect of lesions in adult zebra finches might be due to the relatively stereotyped nature of zebra finch song. Consistent with this possibility, our results demonstrate that lesions of pallial nucleus mMAN affect the more complex and variable song syntax in Bengalese finches. We showed that while mMAN lesions did not abolish the production of song itself or grossly disrupt the acoustic structure of individual syllables, they increased the variability of syllable sequencing; the transition probabilities between syllables became less predictable (increased transition entropy), new branches were introduced into previously stereotyped chunks, and the number of repetitions of repeated syllables (repeat number) became more variable. These results indicate that mMAN is involved in controlling aspects of song syntax in a songbird species that exhibits complex syllable sequencing and suggest that mMAN might play a similar or prominent role in additional species with even greater song complexity, such as canaries or nightingales.

Song in passerine birds relies on a network of recurrently connected nuclei, with different pathways for song production and learning. The song motor pathway consists of primary motor nucleus RA, which projects directly to vocal motor nuclei in the brainstem, and is involved in the moment-by-moment control of the structure of individual syllables (*Vu et al., 1994*; *Sober et al., 2008*; *Miller et al., 2017*). Neurons in premotor song nucleus HVC that project to RA have been shown to encode syllable identity and contribute to syllable timing (*Simpson and Vicario, 1990*; *Vu et al., 1994*; *Hahnloser et al., 2002*; *Long and Fee, 2008*; *Aronov et al., 2011*; *Ölveczky et al., 2011*; *Zhang et al., 2017*). Consistent with a possible role of HVC in also controlling variable syllable sequencing, cooling HVC influences syntax in Bengalese finches (*Zhang et al., 2017*). However, lesions of one input to HVC – the nucleus Nif (*nucleus interfacialis of the anterior nidopallium*) – can also influence syllable sequencing in the Bengalese finch by making the sequence more stereotyped (*Hosino and Okanoya, 2000*; *Cardin et al., 2005*; *Otchy et al., 2015*), raising the question of how much of the control of syntax relies on the internal dynamics of HVC versus activity relayed to HVC by its inputs (*Okanoya and Yamaguchi, 1997*).

In this study, we were especially interested in the possibility that mMAN's inputs to HVC might play an analogous role in modulating syllable sequencing to the role of lMAN's inputs to RA in modulating syllable acoustic structure. In adult birds, lMAN's inputs to RA have been shown to contribute variability to the pitch of individual syllables (*Kao et al., 2005*; *Kao and Brainard, 2006*; *Stepanek*

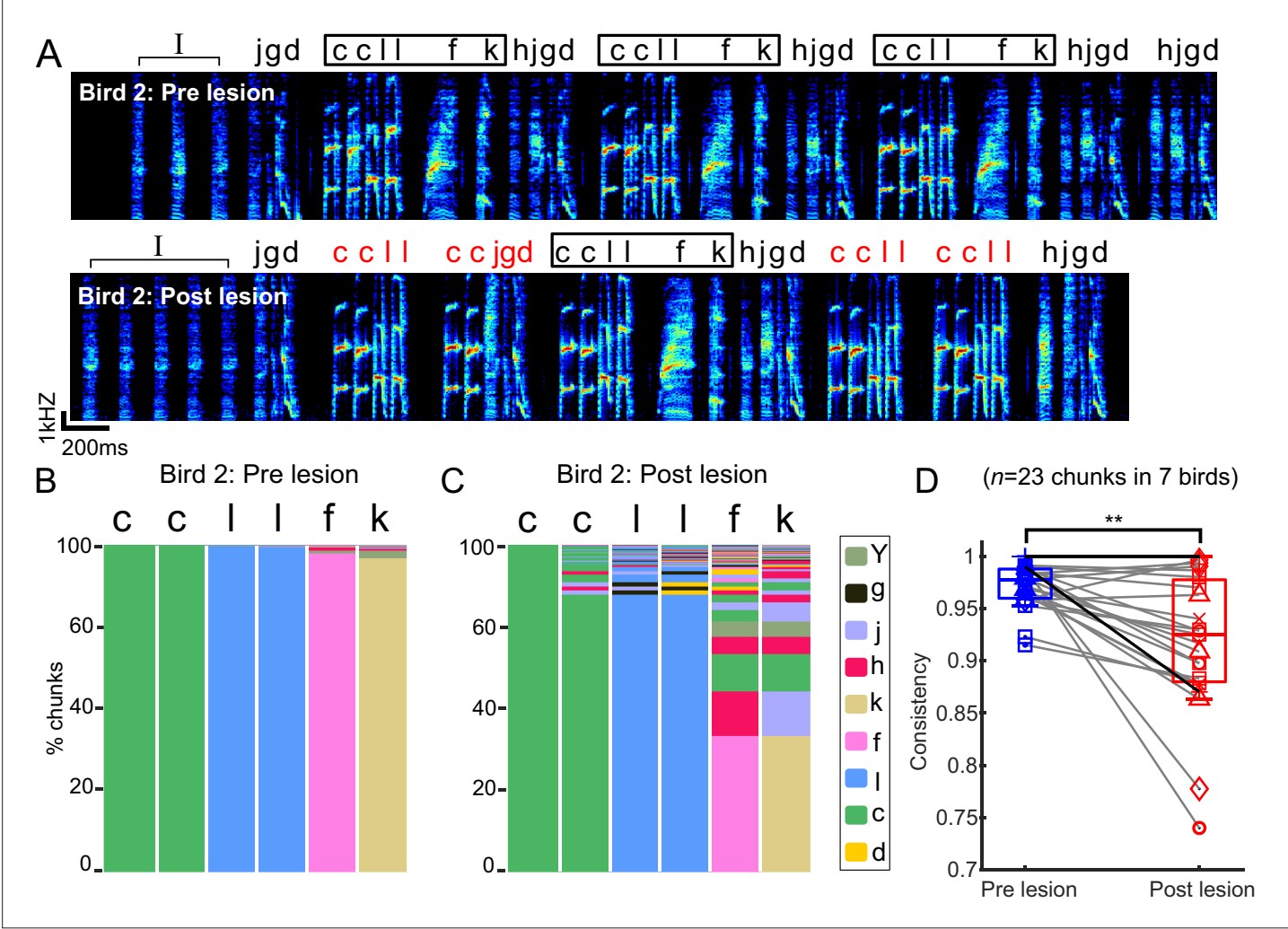

**Figure 3.** Chunks became more variable after bilateral medial magnocellular nucleus of the anterior nidopallium (mMAN) lesions. (**A**) Example spectrogram (bird 2) before and after bilateral mMAN lesions. Atypical chunk sequences are highlighted in red. (**B, C**) Transitions following the first 'c' of the *'ccllfk'* chunk from (**A**) before and after mMAN lesions. Different column colors represent different syllables. (**D**) Chunk consistency before and after bilateral mMAN lesions (\*\*p<0.01, n = 23, Wilcoxon signed-rank test). Example bird is shown as darker line. Boxes show interquartile range and whiskers mark data points within one additional interquartile range.

The online version of this article includes the following source data, source code, and figure supplement(s) for figure 3:

**Source code 1.** Code for generating *Figure 3D* based on data from *Source data 1* - *Source data 28*.

**Source code 2.** Code for generating *Figure 3B and C* based on data from *Source data 1* - *Source data 28*.

**Figure supplement 1.** Change in transition entropy for transitions within chunks vs. branch points.

**Figure supplement 1—source data 1.** Median gap durations for generating *Figure 3—figure supplement 1* for bird 1.

**Figure supplement 1—source data 2.** Median change in transition entropy for generating *Figure 3—figure supplement 1* for bird 1.

**Figure supplement 1—source data 3.** Median gap durations for generating *Figure 3—figure supplement 1* for bird 2.

**Figure supplement 1—source data 4.** Median change in transition entropy for generating *Figure 3—figure supplement 1* for bird 2.

**Figure supplement 1—source data 5.** Median gap durations for generating *Figure 3—figure supplement 1* for bird 3.

**Figure supplement 1—source data 6.** Median change in transition entropy for generating *Figure 3—figure supplement 1* for bird 3.

**Figure supplement 1—source data 7.** Median gap durations for generating *Figure 3—figure supplement 1* for bird 4.

**Figure supplement 1—source data 8.** Median change in transition entropy for generating *Figure 3—figure supplement 1* for bird 4.

**Figure supplement 1—source data 9.** Median gap durations for generating *Figure 3—figure supplement 1* for bird 5.

**Figure supplement 1—source data 10.** Median change in transition entropy for generating *Figure 3—figure supplement 1* for bird 5.

*Figure 3 continued on next page*

*Figure 3 continued*

**Figure supplement 1—source data 11.** Median gap durations for generating *Figure 3—figure supplement 1* for bird 6.

**Figure supplement 1—source data 12.** Median change in transition entropy for generating *Figure 3—figure supplement 1* for bird 6.

**Figure supplement 1—source data 13.** Median gap durations for generating *Figure 3—figure supplement 1* for bird 7.

**Figure supplement 1—source data 14.** Median change in transition entropy for generating *Figure 3—figure supplement 1* for bird 7.

**Figure supplement 1—source code 1.** Code for generating *Figure 3—figure supplement 1* based on data from *Figure 3—figure supplement 1—source data 1*.

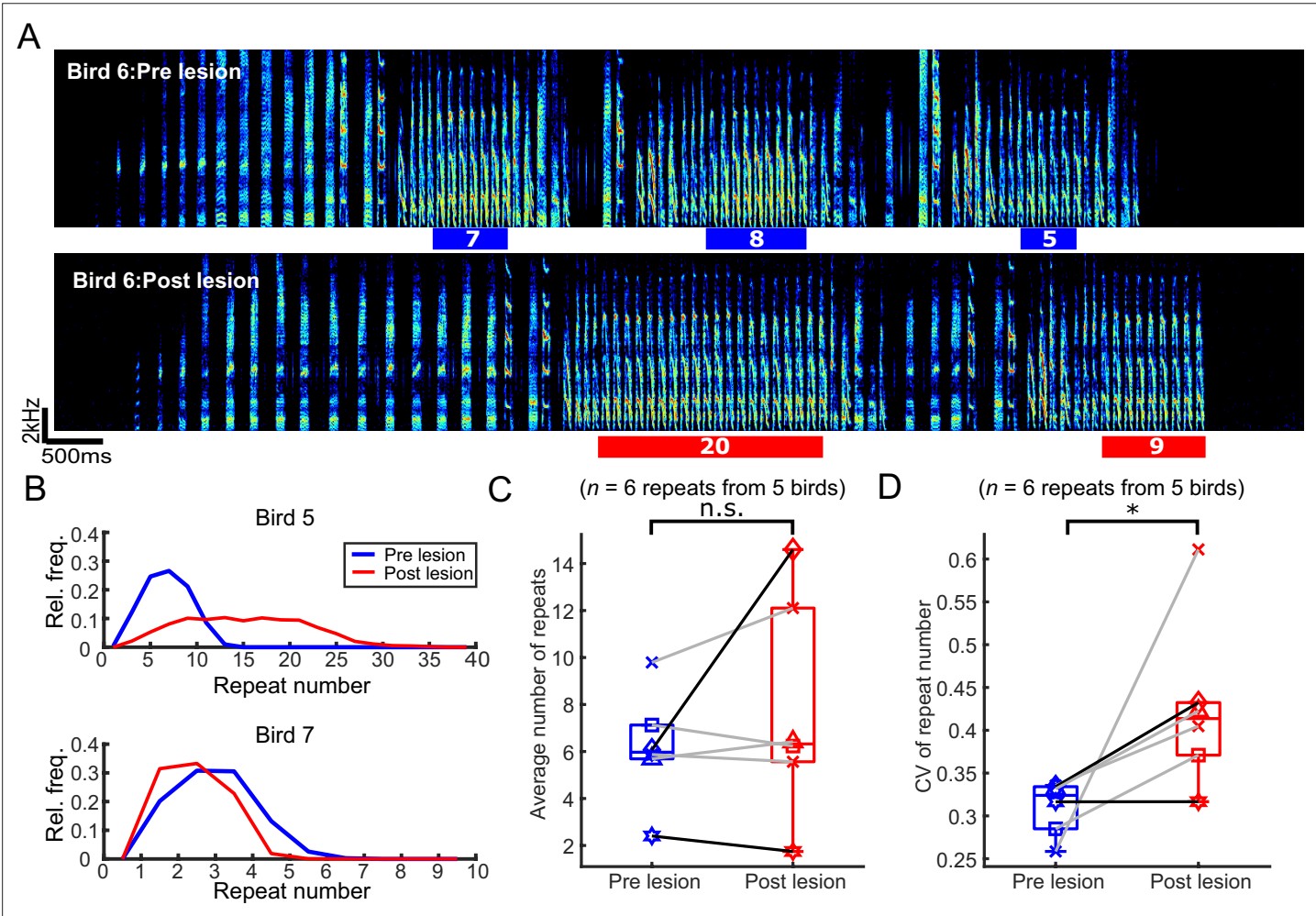

**Figure 4.** The number of syllables per repeat phrase (repeat number) became more variable after bilateral medial magnocellular nucleus of the anterior nidopallium (mMAN) lesions. (**A**) Example spectrogram (bird 6) highlighting one repeat syllable before (blue) and after (red) mMAN lesions. (**B**) Repeat numbers for two additional example birds before (blue) and after (red) mMAN lesions. (**C**) Average repeat numbers before and after mMAN lesions for all repeat phrases (p>0.05, n=6, Wilcoxon signed rank test). Boxes mark interquartile range and whiskers mark data points within one additional interquartile range. (**D**) Coefficient of variation for distribution of repeat numbers before and after mMAN lesions for all repeat phrases (p<0.05, n=6, Wilcoxon signed rank test). Example birds from (**B**) are shown as darker lines. Boxes mark interquartile range and whiskers mark data points within one additional interquartile range.

The online version of this article includes the following source code for figure 4:

**Source code 1.** Code for generating *Figure 4B* based on data from *Source data 1 - Source data 28*.

**Source code 2.** Code for generating *Figure 4D* based on data from *Source data 1 - Source data 28*.

**Source code 3.** Code for generating *Figure 4C* based on data from *Source data 1 - Source data 28*.

and Doupe, 2010), regulate changes to pitch variability between different social contexts (*Kao and Brainard, 2006*; *Hampton et al., 2009*), and bias pitch in the direction of improved performance during learning experiments (*Andalman and Fee, 2009*; *Warren et al., 2011*; *Tian and Brainard, 2017*). The variability contributed by lMAN may facilitate motor exploration that forms the basis of pitch adaptations during learning (*Andalman and Fee, 2009*; *Sober and Brainard, 2012*; *Dhawale et al., 2017*). These effects are mediated by top-down influences of lMAN on RA via a recurrent loop that passes through the songbird basal ganglia (*Kao et al., 2005*; *Andalman and Fee, 2009*; *Kojima et al., 2018*; *Tian et al., 2022*) and are consistent with a role of RA in controlling aspects of syllable acoustic structure. In contrast to these effects on syllable structure, lesions of lMAN in adults have not been found to appreciably influence syllable sequencing (*Hampton et al., 2009*). However, because mMAN forms the output of a parallel recurrent loop that passes through the basal ganglia and projects to HVC, we hypothesized that analogous modulation of syllable sequencing during production and learning of song might be mediated by mMAN. Our finding that lesions of mMAN alter multiple aspects of syllable sequencing is consistent with such an influence on song syntax.

The finding that mMAN can influence song syntax raises the question of whether and how this pathway modulates syllable sequencing in the service of motor exploration, contextual modulation, and learning, in a fashion that is analogous to lMAN's modulation of syllable acoustic structure. That mMAN lesions consistently altered sequence variability indicates a possible role in regulating variability for motor exploration. However, whereas lMAN lesions consistently *decrease* the variability of acoustic structure, we found that mMAN lesions generally *increase* the variability of syllable sequencing, suggesting that top-down contributions to the control of syntax may be different or more distributed than for a syllable feature such as pitch. One possibility is that several inputs to HVC could work synergistically to modulate the syntactical structure of Bengalese finch songs. Consistent with this view, a prior study in Bengalese finches found that lesions of Nif decreased sequence variability (*Hosino and Okanoya, 2000*), suggesting that Nif and mMAN could be playing opposing roles in regulating variability. Our findings also raise the question of whether the medial anterior forebrain pathway, including mMAN, plays a similar role in learning to change syllable sequences during development (*Foster and Bottjer, 2001*; *Lipkind et al., 2013*) or adulthood (*Warren et al., 2012*; *Veit et al., 2021*). In our study, we only recorded song sequencing of male Bengalese finches singing in isolation. Social context, such as female-directed song, can also change song sequencing (*Hampton et al., 2009*; *Chen et al., 2016*). It would be interesting to test whether mMAN plays a role in the social context-modulated changes in sequencing (*Jarvis et al., 1998*), similar to how lMAN contributes to social context-modulated changes in syllable structure (*Sakata et al., 2008*). Further studies with careful manipulations of the basal ganglia-mMAN-HVC circuit will be required to elucidate the mechanism with which this circuit affects sequencing in Bengalese finch song and contributes to these behaviors.

Bengalese finch syllable sequences contain structure at multiple levels, mainly the organization of individual stereotyped chunks, resembling the stereotyped motifs of zebra finch song, and the variable branch points between chunks, repeat phrases, and other individual syllables. One possibility to reconcile findings that both dynamics within HVC and its recurrent inputs affect sequencing could be that the sequence within chunks is encoded predominantly in the synaptic connections of HVC neurons (*Jin, 2009*), while the branching transitions found in Bengalese finch song are influenced by input from other nuclei (*Hosino and Okanoya, 2000*). We here show that mMAN lesions affect both transition entropy at previously variable branch points and transitions within previously stereotyped chunks, indicating that such a simple split of function between mMAN and HVC is not the case.

Besides these, other parts of the song circuit have been shown to affect syllable sequencing. The basal ganglia are involved in selection and sequencing of motor elements in mammals, and manipulations of songbird basal ganglia Area X have previously been shown to affect syllable repetitions in zebra finches and Bengalese finches (*Kobayashi et al., 2001*; *Kubikova et al., 2014*; *Tanaka et al., 2016*; *Xiao et al., 2021*). These effects might be partly mediated via connections through mMAN to HVC (*Kubikova et al., 2007*). Additionally, the control of syllable sequencing must include bilateral coordination of the motor pathways in the two hemispheres (*Schmidt et al., 2004*; *Wang et al., 2008*). As the two hemispheres in birds are not connected by a corpus callosum, the only pathways mediating interhemispheric synchronization of the two HVCs are by bilateral connections through thalamic nuclei. One such pathway involves a projection from thalamic nucleus Uva to HVC (*Coleman*

*and Vu, 2005*; *Hamaguchi et al., 2016*; *Moll et al., 2023*), which is required for song production and contributes to syllable initiation and timing. Another is through thalamic nucleus DMP, which projects bilaterally through mMAN to HVC (*Schmidt et al., 2004*; *Seki and Okanoya, 2008*). mMAN might therefore be well situated to coordinate syllable sequencing in the two hemispheres. Future studies will need to determine whether the effects we observed are the results of removing neuronal processing within mMAN itself or whether mMAN is conveying signals to HVC from other brain areas, including the contralateral hemisphere, which contribute to the control of song syntax.

## Acknowledgements

We thank Jacqueline Göbl, Lioba Fortkord, Yarden Cohen, Tim Gardner, and Felix Moll for helpful discussions and comments on earlier versions of this manuscript. We thank David Nicholson, Jonathan Wong, and Peter Pilz for help with technical questions. Michael Brainard was supported by the Howard Hughes Medical Institute. Lena Veit was supported as a Howard Hughes Medical Institute Fellow of the Life Sciences Research Foundation and by a Daimler Benz postdoctoral fellowship. Avani Koparkar was supported by the Studienstiftung des deutschen Volkes and International Max Planck Research School for Mechanisms of Mental Function and Dysfunction (IMPRS-MMFD). Jonathan Charlesworth and Timothy Warren were supported by the National Science Foundation (NSF) Graduate Research Fellowship. We acknowledge support from the Open Access Publication Fund of the University of Tübingen.

## Additional information

### Competing interests

Jonathan D Charlesworth: Jonathan Charlesworth is employed at Noctrix Health Inc and declares no related competing interests. Sooyoon Shin: Sooyoon Shin reports employment at Verily Life Sciences and equity ownership in Verily Life Sciences. The other authors declare that no competing interests exist.

### Funding

| Funder | Grant reference number | Author |
| --- | --- | --- |
| Howard Hughes Medical Institute | | Michael S Brainard |
| Life Sciences Research Foundation | | Lena Veit |
| Daimler Benz Postdoctoral Fellowship | | Lena Veit |
| Studienstiftung des Deutschen Volkes | | Avani Koparkar |
| National Science Foundation Graduate Research Fellowship Program | | Timothy L Warren Jonathan D Charlesworth |

The funders had no role in study design, data collection and interpretation, or the decision to submit the work for publication.

### Author contributions

Avani Koparkar, Data curation, Software, Formal analysis, Funding acquisition, Visualization, Writing – original draft, Writing – review and editing; Timothy L Warren, Jonathan D Charlesworth, Funding acquisition, Investigation, Methodology, Writing – review and editing; Sooyoon Shin, Investigation, Methodology, Writing – review and editing; Michael S Brainard, Conceptualization, Resources, Supervision, Funding acquisition, Writing – review and editing; Lena Veit, Conceptualization, Resources, Software, Supervision, Funding acquisition, Writing – original draft, Writing – review and editing

## Author ORCIDs

Avani Koparkar (iD) https://orcid.org/0009-0004-3462-9453
Timothy L Warren (iD) http://orcid.org/0000-0002-4429-4106
Jonathan D Charlesworth (iD) https://orcid.org/0000-0002-6651-0009
Sooyoon Shin (iD) http://orcid.org/0000-0002-0339-4856
Michael S Brainard (iD) http://orcid.org/0000-0002-9425-9907
Lena Veit (iD) http://orcid.org/0000-0002-9566-5253

## Ethics

All experimental work was carried out under protocols approved by the University of California, San Francisco (UCSF)'s institutional animal care and use committee (IACUC); protocol entitled "The neural basis of vocal learning in songbirds," including versions AN185512, AN170723, AN107972, AN087315, AN080388, AN075783.

Reviewer #1 (Public Review): https://doi.org/10.7554/eLife.93272.3.sa1
Reviewer #2 (Public Review): https://doi.org/10.7554/eLife.93272.3.sa2
Author response https://doi.org/10.7554/eLife.93272.3.sa3

---

# Additional files

## Supplementary files

• MDAR checklist

• Source data 1. Sequence data for bird 1 post lesion with introductory notes and repeats replaced.

• Source data 2. Sequence data for bird 1 post lesion with only introductory notes replaced.

• Source data 3. Sequence data for bird 1 pre lesion with introductory notes and repeats replaced.

• Source data 4. Sequence data for bird 1 pre lesion with only introductory notes replaced.

• Source data 5. Sequence data for bird 2 post lesion with introductory notes and repeats replaced.

• Source data 6. Sequence data for bird 2 post lesion with only introductory notes replaced.

• Source data 7. Sequence data for bird 2 pre lesion with introductory notes and repeats replaced.

• Source data 8. Sequence data for bird 2 pre lesion with only introductory notes replaced.

• Source data 9. Sequence data for bird 3 post lesion with introductory notes and repeats replaced.

• Source data 10. Sequence data for bird 3 post lesion with only introductory notes replaced.

• Source data 11. Sequence data for bird 3 pre lesion with introductory notes and repeats replaced.

• Source data 12. Sequence data for bird 3 pre lesion with only introductory notes replaced.

• Source data 13. Sequence data for bird 4 post lesion with introductory notes and repeats replaced.

• Source data 14. Sequence data for bird 4 post lesion with only introductory notes replaced.

• Source data 15. Sequence data for bird 4 pre lesion with introductory notes and repeats replaced.

• Source data 16. Sequence data for bird 4 pre lesion with only introductory notes replaced.

• Source data 17. Sequence data for bird 5 post lesion with introductory notes and repeats replaced.

• Source data 18. Sequence data for bird 5 post lesion with only introductory notes replaced.

• Source data 19. Sequence data for bird 5 pre lesion with introductory notes and repeats replaced.

• Source data 20. Sequence data for bird 5 pre lesion with only introductory notes replaced.

• Source data 21. Sequence data for bird 6 post lesion with introductory notes and repeats replaced.

• Source data 22. Sequence data for bird 6 post lesion with only introductory notes replaced.

• Source data 23. Sequence data for bird 6 pre lesion with introductory notes and repeats replaced.

• Source data 24. Sequence data for bird 6 pre lesion with only introductory notes replaced.

• Source data 25. Sequence data for bird 7 post lesion with introductory notes and repeats replaced.

• Source data 26. Sequence data for bird 7 post lesion with only introductory notes replaced.

• Source data 27. Sequence data for bird 7 pre lesion with introductory notes and repeats replaced.

• Source data 28. Sequence data for bird 7 pre lesion with only introductory notes replaced.

## Data availability

All data are present in the article and/or the Supplementary Materials, and Code is shared with the article and at: https://github.com/avanikop/mMAN_lesions (copy archived at *avanikop, 2024*).

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
