## [Editor Report · eLife assessment]

Songbirds provide a tractable model system to study mechanisms of vocal production and sequencing, and past work showed that lesions to lMAN, the output of a basal ganglia thalamocortical loop, reduced vocal variability, consistent with a role in motor exploration. In this **fundamental** work, the authors rigorously examined how lesions to an understudied neighboring region, mMAN, part of a parallel basal ganglia loop, affect singing in Bengalese finches, whose songs exhibit complex sequential transitions. The authors provide **compelling** evidence that mMAN lesions resulted in increased sequential variability but do not affect syllable acoustic structure, showing that distinct frontal systems can have distinct functions for producing and sequencing song syllables.

---

## [Referee Report · Reviewer #1 (Public Review)]

Summary:

Songbirds provide a tractable system to examine neural mechanisms of sequence generation and variability. In past work, the projection from LMAN to RA (output of the anterior forebrain pathway) was shown to be critical for driving vocal variability during babbling, learning, and adulthood. LMAN is immediately adjacent to MMAN, which projects to HVC. MMAN is less well understood but, anatomically, appears to resemble LMAN in that it is the cortical output of a BG-thalamocortical loop. Because it projects to HVC, a major sequence generator for both syllable phonology and sequence, a strong prediction would be that MMAN drives sequence variability in the same way that LMAN drives phonological variability. This hypothesis predicts that MMAN lesions in a Bengalese finch would reduce sequence variability. Here, the authors test this hypothesis. They provide a surprising and important result that is well motivated and well analyzed: MMAN lesions increase sequence variability - this is exactly the opposite result from what would be predicted based on the functions of LMAN.

Strengths:

(1) A very important and surprising result shows that lesions of a frontal projection from MMAN to HVC, a sequence generator for birdsong, increase syntactical variability.

(2) The choice of Bengalese finches, which have complex transition structures, to examine the mechanisms of sequence generation, enabled this important discovery.

(3) The idea that frontal outputs of BG-cortical loops can generate vocal variability comes from lesions/inactivations of a parallel pathway from LMAN to RA. The difference between MMAN and LMAN functions is striking and important.

---

## [Referee Report · Reviewer #2 (Public Review)]

Summary:

This study investigates the neural substrates of syntax variation in Bengalese finch song. Here, the authors tested the effects of bilateral lesions of mMAN, a brain area with inputs to HVC, a premotor area required for song production. Lesions in mMAN induce variability in syntactic elements of song specifically through increased transition entropy, variability within stereotyped song elements known as chunks and increases in the repeat number of individual syllables. These results suggest that mMAN projections to HVC contribute to multiple aspects of song syntax in the Bengalese finch. Overall the experiments are well-designed, the analysis excellent, and the results are of high interest.

Strengths:

The study identifies a novel role for mMAN, medial magnocellular nucleus of the anterior nidopallium, in the control of syntactic variation within adult Bengalese finch song. This is of particular interest as multiple studies previously demonstrated that mMAN lesions to do not effect song structure in zebra finches. The study undertakes a thorough analysis to characterise specific aspects of variability within the song of lesioned animals. The conclusions are well supported by the data.

---

## [Author Response]

The following is the authors’ response to the original reviews.

**Public reviews:**

**Reviewer #1 (Public Review):**
Summary:Songbirds provide a tractable system to examine neural mechanisms of sequence generation and variability. In past work, the projection from LMAN to RA (output of the anterior forebrain pathway) was shown to be critical for driving vocal variability during babbling, learning, and adulthood. LMAN is immediately adjacent to MMAN, which projects to HVC. MMAN is less well understood but, anatomically, appears to resemble LMAN in that it is the cortical output of a BG-thalamocortical loop. Because it projects to HVC, a major sequence generator for both syllable phonology and sequence, a strong prediction would be that MMAN drives sequence variability in the same way that LMAN drives phonological variability. This hypothesis predicts that MMAN lesions in a Bengalese finch would reduce sequence variability. Here, the authors test this hypothesis. They provide a surprising and important result that is well motivated and well analyzed: MMAN lesions increase sequence variability - this is exactly the opposite result from what would be predicted based on the functions of LMAN.Strengths:(1) A very important and surprising result shows that lesions of a frontal projection from MMAN to HVC, a sequence generator for birdsong, increase syntactical variability.(2) The choice of Bengalese finches, which have complex transition structures, to examine the mechanisms of sequence generation, enabled this important discovery.(3) The idea that frontal outputs of BG-cortical loops can generate vocal variability comes from lesions/inactivations of a parallel pathway from LMAN to RA. The difference between MMAN and LMAN functions is striking and important.Weaknesses:(1) If more attention was paid to how syllable phonology was (or was not) affected by MMAN lesions then the claims could be stronger around the specific effects on sequence.
**Reviewer #2 (Public Review):**
Summary:This study investigates the neural substrates of syntax variation in Bengalese finch songs. Here, the authors tested the effects of bilateral lesions of mMAN, a brain area with inputs to HVC, a premotor area required for song production. Lesions in mMAN induce variability in syntactic elements of song specifically through increased transition entropy, variability within stereotyped song elements known as chunks, and increases in the repeat number of individual syllables. These results suggest that mMAN projections to HVC contribute to multiple aspects of song syntax in the Bengalese finch. Overall the experiments are well-designed, the analysis excellent, and the results are of high interest.Strengths:The study identifies a novel role for mMAN, the medial magnocellular nucleus of the anterior nidopallium, in the control of syntactic variation within adult Bengalese finch song. This is of particular interest as multiple studies previously demonstrated that mMAN lesions do not affect song structure in zebra finches. The study undertakes a thorough analysis to characterise specific aspects of variability within the song of lesioned animals. The conclusions are well supported by the data.Weaknesses:The study would benefit from additional mechanistic information. A more fine-grained or reversible manipulation, such as brain cooling, might allow additional insights into how mMAN influences specific aspects of syntax structure. Are repeat number increases and transition entropy resulting from shared mechanisms within mMAN, or perhaps arising from differential output to downstream pathways (i.e. projections to HVC)? Similarly, unilateral manipulations would allow the authors to further test the hypothesis that mMAN is involved in inter-hemispheric synchronization.

We thank the reviewers and editor for their encouraging and helpful comments and suggestions. We have revised the previous submission with new analyses and discussion to address points raised by the reviewers.

Following the suggestion of Reviewer 1 we have added an analysis of the effects of mMAN lesions on syllable phonology, using a variety of measures. We have included 3 new Figure Supplements that detail our analyses and elaborate on these points.

We agree with Reviewer 2 that reversible and unilateral manipulations would be interesting and potentially enable additional insights into the mechanisms by which mMAN influences song sequencing, and we are planning to perform such experiments in future studies.

We made additional minor changes throughout the manuscript to address other points raised by the reviewers, and we thank them again for their time and effort in providing constructive feedback to improve our study.

A complete point by point detailing of these changes is included below, interspersed with the reviewer comments.

**Reviewer #1 (Recommenda1ons For The Authors):**
The opposite result from what would be predicted based on the functions of LMAN.Shoring up the paper's claims and ruling out alternative interpretations will require attention to the following issues:Major comments(1) Acoustic structure of syllablesLine 294 & Sup. Figure 2, in some birds the syllable acoustic structures seem to be significantly different between the pre- and post-lesion condition, e.g. 'w' in Bird 1, 'g' in Bird 2, 'blm' in Bird 6. This observation seems to contradict the claim that acoustic structures are not affected by MMAN lesions.Related to the previous point, a more detailed analysis is needed to quantify the extent of acoustic changes caused by MMAN lesions. For example, do these pre- and post- lesion syllables form distinct clusters if embedded in a UMAP? Do more standard measures of syllable phonology (e.g. SAP similarity scores or feature distributions) show differences in pre- and post-MMAN lesion?

We agree with the reviewer that there were individual syllables as illustrated in the average spectrograms of Figure 2 – figure supplement 1 that qualitatively differed between pre- and post-lesion recordings. We have followed the reviewer’s suggestion to quantify changes to syllable phonology using both similarity scores by Sound Analysis Pro (SAP) and a variety of identified acoustic features.

In brief, these measures largely corroborate the conclusion that for most birds and syllables there was little or no difference in phonology between pre- and post-lesion songs, but that in a minority of cases syllables were altered noticeably (further detail below). In those cases where syllable phonology was altered, changes were not consistent across birds, and we cannot rule out off-target effects due to damage to structures or fibers of passage neighboring mMAN, so that it is unclear whether some subtle changes to syllable phonology can be attributed to mMAN lesions versus other causes. Future studies could more specifically examine whether damage to mMAN alone is sufficient in some cases to degrade syllable structure by using viral or other approaches that enable the more specific disruption of mMAN projection neurons.

In practice, almost all syllables were identifiable in post-lesion songs so that we could unambiguously assign identity for purposes of evaluating effects of lesions on sequencing. Moreover, in any individual cases where there was ambiguity in syllable identity, we used the sequential context to assign the most likely label. Thus, any errors in assignment in such cases would have tended to reduce rather than accentuate the magnitude of reported sequencing effects. Lastly, each of the reported effects of mMAN lesions on sequencing were observed in multiple birds for which we detected no significant changes to syllable similarity.

Further details of the analyses of syllable structure are detailed below, and have been added as new figure supplements:

(1) Syllable similarity scores calculated using SAP (Sound Analysis Pro) (new Figure 2 – figure supplement 2). We compared pre-post lesion similarity scores for each syllable with self-similarity measures for the same syllables taken from separate control recordings before lesions. For comparison, we also included a cross-similarity score for syllables of different types. These measures confirmed the qualitative impression from spectrograms that for most birds there were no greater changes to syllable structure following lesions than was present across control recordings. For one bird, pre-post changes were significantly larger than changes across control recordings, but pre-post similarity remained higher than cross-similarity.

(2) Analysis of fundamental frequency and coefficient of variation (CV) of fundamental frequency of select syllables for each bird before and after mMAN lesions (new Figure 2- figure supplement 3). This analysis is directly comparable with the same analysis performed on LMAN lesions in Sakata, Hampton, Brainard (2008). We carried out this analysis in part to address changes to syllable structure that might have inadvertently arisen due to damage to LMAN, which sits immediately lateral to mMAN. In the Bengalese finch and zebra finch, lesions of LMAN cause little change to the mean fundamental frequency of individual syllables but cause a consistent reduction in the coefficient of variation (CV) of fundamental frequency across repeated renditions of a given syllable (Sakata, Hampton, Brainard 2008, Andalman, Fee 2009, Warren et al. 2011,). We therefore supposed that unintended damage to LMAN or its projections to RA might have resulted in a reduction in the CV of syllables following mMAN lesions. Instead, we saw a modest increase in the CV of fundamental frequency (mean across birds of +20%; range -19 to +43%). These data suggest that off target effects on LMAN were largely absent in our experiments (consistent with histology, e.g. Figure 1 - figure supplement 1).

(3) Comparison of Entropy of spectral envelope (entS), Temporal centroid for the temporal envelope (meanT), First, second and third formants (F1, F2, F3), before and after lesions calculated using the python SoundSig toolbox (Elie and Theunissen 2016) (new Figure 2- figure supplement 4). Acoustic features generally showed little change between pre and post lesion songs. They highlight as relative outliers the same individual examples that stand out in the average spectrograms in Figure 2 – figure supplement 1.

**Author response image 1. sa3fig1:** Syllable similarity calculated using Sound Analysis Pro (SAP). ‘Self Similarity’ = Similarity comparison of syllables before mMAN lesions to syllables of the same type, taken from two separate control recordings before the lesions, ‘Pre vs Post’ = Similarity comparison of the same syllable types before and after mMAN lesions, ‘Cross Similarity’ = Similarity comparison of each syllable type to other syllable types. For Birds 1-2 and 4-7, ‘Self Similarity’ was not significantly different from ‘Pre vs Post’ Similarity (p>0.05, Wilcoxon sign rank test), while for Bird 3, there was a significant difference (p = 0.03, Wilcoxon sign rank test). For all birds ‘Pre vs Post’ was significantly different from ‘Cross Similarity’ (p<0.05, Wilcoxon sign rank test). On average, ‘Pre vs Post’ was 4.8 % less than ‘Self Similarity’ (range 0.2%-14%) while ‘Cross Similarity’ was 40% less than ‘Self Similarity’ (range 20.2%-56.3%). These measures confirm the qualitative impression from Figure 2- figure supplement 1 that for most birds and syllables there were no greater changes to syllable structure following lesions than was present across control recordings, and that pre-post similarity remained higher than cross-similarity, i.e. syllables remained clearly identifiable.

**Author response image 2. sa3fig2:** (A) CV of fundamental frequency (FF) of select syllables before and after mMAN lesions. In the Bengalese finch and zebra finch, lesions of lMAN, which sits immediately lateral to mMAN, cause a consistent reduction in the coefficient of variation (CV) of fundamental frequency across repeated renditions of a given syllable (Sakata, Hampton, Brainard 2008, Andalman, Fee 2009, Warren et al. 2011). We therefore supposed that unintended damage to lMAN or its projections to RA might have resulted in a reduction in the CV of syllables following mMAN lesions. Instead we saw a modest increase in the CV of fundamental frequency (p<0.05, Wilcoxon sign rank test; mean across birds of +20%; range -19 to +43%). These data suggest that it is unlikely that changes to syllable structure might have arisen due to accidental damage to lMAN. (B) Percent change in mean fundamental frequency aqer mMAN lesions vs mean fundamental frequency before mMAN lesions.

**Author response image 3. sa3fig3:** Selected acoustic features for all syllables in all birds before and after mMAN lesions. Different colors represent different syllable types per bird. ‘entS’ = Entropy of spectral envelope, ‘meanT’ = Temporal centroid for temporal envelope, ‘F1’ = First formant, ‘F2’ = Second formant, ‘F3’ = Third formant. Acoustic features generally showed little change between pre and post lesion songs. They highlight as relative outliers the same individual examples that stand out in the average spectrograms in Figure 2 – figure supplement 1.

(2) Shoring up claims of increased transitional variabilityLine 301 & Sup. Figure 1, in several birds (1, 2, 5, 6), seems that there is a downward trend for postlesion, i.e. the transition entropy gradually decreases with time. How to exclude the possibility that the increased variability is a transient effect, e.g. caused by surgery side effects or destabilization of circuits, which may eventually recover to normal?

Transition entropy remains elevated for as long as the birds were followed in this study. While the persistence of the effects we observed is longer than transient effects such as those following Nif lesion in zebra finches (Otchy et al., 2015 ~2 days), we cannot rule out either recovery or further deterioration following lesions on much longer time scales, such as those reported by Kubikova et al., 2007 (X lesions, 6 months). We have now added data points for 4 birds where we had songs from later timepoints following lesions; for three of these birds, transition entropy remained elevated above the baseline values for 14 and 33 days, respectively (Figure 1 - figure supplement 2).

Line 313 & Sup. Figure 4, the claim that "transitions that had low history dependence tended to show larger changes after mMAN lesions" needs better statistical support, because in Sup. Figure 4, the correlation is not significant.

We apologize for the phrasing. We have changed the sentence to: “Consistent with the first possibility, we observed that there was a nonsignificant trend toward larger changes after mMAN lesion for transitions with low history dependence.”

Figure 4C-D, only data from 5 out of 7 birds was included, did the other two birds not have repeats? If so, the authors need to be explicit on data exclusion.

The reviewer’s inference is correct that in our dataset only 5 out of 7 birds had songs which contained repeat phrases. We have added the following sentence to state that explicitly: “In our dataset of 7 birds, only 5 birds had songs which contained repeat phrases.”

Minor commentsSup. Figure 3, to help readers understand, (1) add symbols and arrows to point to the structures; (2) indicate the orientation of the slide, e.g. which direction is medial/lateral; (3) a negative control without lesion needs to be shown for comparison.

We have made the suggested changes and updated new Figure 1- figure supplement 1.

**Author response image 4. sa3fig4:** Image of calcitonin gene-related peptide (CGRP)-stained frontal section (left) control and (right) bird 5. CGRP labels cells in both lMAN (seen in black to the left of the lesion) and mMAN (blue, intact; red, completely destroyed).

A statistical test is needed for Sup. Figure 5B.

We have modified the Figure legend for Figure 3 – figure supplement 1 as follows:

“Change in transition entropy was not significantly different for transitions within chunks and at branchpoints (p> 0.05, Wilcoxon rank sum test)”

Line 363, these can be moved to the Introduction, so readers have a better sense of what's already known about MMAN lesion.

We have moved the sentence to Introduction.

Fig 1e. RA also projects to DLM.

Our intention was to focus on the connections involving mMAN; we have now added the connection in Figure 1E.

**Reviewer #2 (Recommenda1ons For The Authors):**
Please address this issue in the discussion (no new experiments required): It would be interesting to consider how social context modulates the variability of the song. In these experiments, Bengalese finches were singing in isolation. How might changes in syntax be modulated by the presence of a female in directed song and in other social contexts?

Thank you for your suggestion. One study by Jarvis, et al., (Jarvis E., et al., 1998) shows that ZENK expression in mMAN after singing does not differ between female-directed singing, undirected singing and singing in presence of a male conspecific. This suggests that activity in mMAN might not be modulated by social context. But we agree that it would be interesting to test how a change in social context (which typically leads to reduced transition entropy) interacts with the increased variability we see after mMAN lesions. We have added the following sentences to the discussion:

“In our study, we only recorded song sequencing of male Bengalese finches singing in isolation. Social context, such as female-directed song, can also change song sequencing (Hampton, Sakata and Brainard, 2009; Chen, Matheson and Sakata, 2016). It would be interesting to test whether mMAN plays a role in the social context-modulated changes in sequencing (Jarvis et al., 1998), similar to how lMAN contributes to social context-modulated changes in syllable structure (Sakata, Hampton and Brainard, 2008).”